# The Petunia CHANEL Gene is a ZEITLUPE Ortholog Coordinating Growth and Scent Profiles

**DOI:** 10.3390/cells8040343

**Published:** 2019-04-11

**Authors:** Marta I. Terry, Fernando Pérez-Sanz, M. Victoria Díaz-Galián, Felipe Pérez de los Cobos, Pedro J. Navarro, Marcos Egea-Cortines, Julia Weiss

**Affiliations:** 1Genética Molecular, Instituto de Biotecnología Vegetal, Edificio I+D+I, Plaza del Hospital s/n, Universidad Politécnica de Cartagena, 30202 Cartagena, Spain; marta.terry@edu.upct.es (M.I.T.); mariavictoria.diaz@edu.upct.es (M.V.D.-G.); Marcos.Egea@upct.es (M.E.-C.); 2Biomedical Informatic and Bioinformatic Platform, Biomedical Research Institute of Murcia, University Clinical Hospital ‘Virgen de la Arrixaca’, University of Murcia, 30120 Murcia, Spain; fernando.perez8@um.es; 3Plant Breeding Department, Center of Edafology and Applied Biology of Segura-High Council for Scientific Research (CEBAS-CSIC), Espinardo University Campus, Espinardo, 30100 Murcia, Spain; fpcobos@cebas.csic.es; 4Escuela Técnica Superior de Ingeniería de Telecomunicación (DSIE), Campus Muralla del Mar, s/n., Universidad Politécnica de Cartagena, 30202 Cartagena, Spain; pedroj.navarro@upct.es

**Keywords:** floral scent, petal development, growth rate, phenomics, circadian clock, ZEITLUPE, image analysis, hairpin RNA

## Abstract

The floral perianth, comprising sepals and petals, conceals the sexual organs and attracts pollinators. The coordination of growth and scent emission is not fully understood. We have analyzed the effect of knocking down *CHANEL* (*PhCHL*), the *ZEITLUPE* ortholog in petunia (*PhCHL*) by hairpin RNAs. Plants with low *PhCHL* mRNA had overall decreased size. Growth evaluation using time lapse image analysis showed that early leaf movement was not affected by *RNAi:PhCHL*, but flower angle movement was modified, moving earlier during the day in knockdown plants than in wild types. Despite differences in stem length, growth rate was not significantly affected by loss of *PhCHL*. In contrast, petal growth displayed lower growth rate in *RNAi:PhCHL*. Decreased levels of *PhCHL* caused strongly modified scent profiles, including changes in composition and timing of emission resulting in volatile profiles highly divergent from the wild type. Our results show a role of *PhCHL* in controlling growth and development of vegetative and reproductive organs in petunia. The different effects of *PhCHL* on organ development indicate an organ-specific interpretation of the down regulation of *PhCHL*. Through the control of both timing and quantitative volatile emissions, *PhCHL* appears to be a major coordinator of scent profiles.

## 1. Introduction

Plant aerial organs grow from lateral primordia that form in the shoot apical meristem [1]. The type of organs produced, i.e., leaves or flowers, are the result of a vegetative or reproductive developmental program. The formation of flowers is the result of the activation of the so-called floral organ identity genes. They comprise a set of MADS-BOX proteins that in a combinatorial fashion allow the formation of the different organs [2]. The interaction of different MADS-box proteins occur via formation of protein complexes that activate the different organ identity programs leading to the formation of sepals and petals in the perianth and stamens and carpels [3,4].

Floral organs play an important role in reproductive success in many plants. Proper floral size is a key component of flower-pollinator interaction [5]. Floral size, like in leaves, is controlled by coordinated cell division and expansion processes [6]. The genes controlling lateral organ size appear to be conserved. Indeed, general regulators of lateral growth such as *AINTEGUMENTA* control floral size in Arabidopsis, petunia, *Antirrhinum* and tobacco [7,8,9].

During late flower development and maturation there is a major transcriptional reprogramming and scent emission starts with flower opening [10,11]. Scent emission takes place when the floral organ identity genes are not fully active, indicating a quantitative effect of the DEFICIENS/GLOBOSA MADS box proteins and downstream factors on scent emission [11].

There are a large number of plants that emit floral volatiles with significantly larger outputs during the day or during the night [12,13,14,15,16], suggesting a circadian regulation of scent emission. The circadian clock genes *LATE ELONGATED HYPOCOTYL* in petunia (*PhLHY*) and *NaLHY* and *ZEITLUPE* (*NaZTL*) in *Nicotiana attenuata* have been investigated and control the timing of emission of methyl benzoate and benzyl benzoate in petunia and benzyl acetone in *N. attenuata* [17,18]. Thus, a default pathway controlled by petal identity may activate floral scent emission, and the fine tuning in terms of emission timing should be regulated by the clock. However, this emerging hypothesis requires further experimental support; the studies mentioned have analyzed a small number of volatile organic compounds (VOCs), but the effect, if any, of the circadian clock on scent profiles is not known.

In this work, we have analyzed the down-regulation of the petunia gene ortholog of *ZTL* and its effect on growth and scent emission. We found that *PhZTL* plays a differential role in stem and floral size and is a major coordinator of floral scent profiles in petunia. As a result, we named the gene *CHANEL* (*PhCHL*).

## 2. Materials and Methods

### 2.1. Gene Identification and Phylogenetic Analysis

We identified *ZTL* orthologs and paralogs in the Petunia genome using BLAST [19]. The identified scaffolds and cDNAs were used to confirm the genome annotation using Genewise [20]. Protein alignment was performed with CLUSTALX [21]. Phylogenetic analysis was performed with the R libraries ape and phangorn [22,23], using the Maximum Likelihood as statistical method, JTT (Jones, Taylor and Thornton) as model of amino acid substitution, and 1000 bootstrap replicates. Trees were visualized and annotated with ggtree [24] using R, (R version 3.5.1). Protein accessions are listed in Appendix A.

### 2.2. Silencing of PhCHL

We obtained sequence information from the genomic clone of *PhCHL* (Peaxi162Scf01124g00126.1) found in W115 (or Mitchell) and amplified 255 bp encompassing the last nine coding codons and 3′untranslated region. As the major effect of silencing of *ZTL* in petunia was a major disruption of scent profiles, we called it *CHANEL*, a famous perfume. The specific sequence of *PhCHL* was:

>PhCHL

TGAACTATCTTTAGCAAGCTCTGTCATTTGAATAAAGAAAAAAGTAATGATGAAGAGAAGGTGTTGTGCAGTATTCATAATGAAAATTTTGCCTCAAGAATAAAGAGAGTCCCGAGCAAACTATTGCAGTGCGGTTTTTGCATTGCACCAAATGCATAAATGACTAGCAAGTACCTGTGAGTTAGTGGCTGTCTTGTTTATTCTTGTGTGGCTCATATGCCATGGTGAGCAAATGGTCCTATTGAGCAGATGGTC 

We used primers, introducing a partial attb1 and attb2 recombination site for GATEWAY cloning using PhCHLrnaiaattb1 and PhCHLrnaiattb2 (Appendix A). We performed a second PCR using attb1 and attb2 to add the corresponding sequences for recombination. The amplified fragment was recombined into pDONR221 and into the GATEWAY vector pHELLSGATE12 to obtain a hairpin construct [25]. The pHELLSGATE12 drives the hairpin construct using a standard 35S promoter. The W115 Mitchell double haploid was transformed as described previously [7].

Transformed plants were identified by PCR with primers amplifying NPTII. The completeness of the construct was tested with the Agri 51 and Agri 56 primers [26] (see Appendix A). The PCR conditions were 3 min at 95 °C followed by 35 cycles of 15 s at 95 °C, 15 s at 55 °C and 15 s at 72 °C, and terminated by 5 min at 72 °C (Kapa Biosistems). The PCR reactions were loaded on 1% agarose gel containing ethydium bromide. PCR products from T2 plants were purified and sequenced.

### 2.3. Plant Growth Conditions and Sampling

Petunia plants were grown using a commercial substrate in a greenhouse under natural conditions. All experiments were conducted with at least three biological replicates. To study the expression of *PhCHL*, non-transgenic siblings and two independent *RNAi:PhCHL* T2 lines (*RNAi:PhCHL3* and *RNAi:PhCHL10*) were transferred to a climate chamber with 12 h of light and 12 h of dark (12LD) or 8L:16D and 23 °C and 18 °C for day and night, respectively. Plants were acclimated for 4–5 days. Young leaves and petals from 2–3 day-old flowers were sampled every three h for 24 h; tissues were immediately frozen in liquid nitrogen and stored at -80 °C until further analysis.

To analyze the floral scent of non-transgenic and transgenic petunia flowers, we took at least three biological replicates of 2–3 day-old flowers (one flower per plant) every three h.

### 2.4. Housekeeping Genes and Gene Expression Analysis by qPCR

RNA was extracted from three biological replicates per time point of leaves and corollas using acid phenol [27]. Concentrations were measured using NanoDrop (Thermo-Fisher). Equal amounts of total RNA were used to obtain cDNA using Maxima kits (Thermo-Fisher) according to the user manual.

Previously, we performed a study to validate the housekeeping gene or genes, in two tissue samples, petal and leaf, for time course studies according to [28]. The candidate genes were *ACTIN 11* (*ACT*), *CYCLOPHILIN* (*CYP*), *ELONGATION FACTOR 1α* (*EF1α*), *GLYCALDEHYDE-3-PHOSPHATE DEHYDROGENASE* (*GADPH*), *RIBOSOMAL PROTEIN S13* (*RPS13*), *GTP-NUCLEAR BINDING PROTEIN* (*RAN1*) and *POLYUBIQUITIN* (*UBQ*) (Appendix A). PCR analysis was performed as described previously [28]. The following protocol was used for 40 cycles: 95 °C for 5 s, 60 °C for 20 s and 72 °C for 15 s (Clontech SYBR Green Master Mix and Mx3000P qPCR Systems, Agilent Technologies), samples were run in duplicate. Cycle threshold (Ct) values were analyzed by BestKeeper [29], NormFinder [30], geNorm [31] and comparative ΔCt methods [32] implemented in the web-based tool RefFinder [33] at different time points (Appendix A).

Analyzing all tissues, the most stable genes were *PhCYP* and *PhEF1α* (Appendix A). Moreover, for individual tissues, results were slightly different: the best genes for normalization in leaves were *PhACT* and *PhEF1α*, and for petals, *PhACT* and *PhCYP* (Appendix A).

We used *PhACT* to normalize the expression of clock genes in petunia leaves and petals as described [34]. Normalized expression was calculated as described [34] using the REST program [29] and *PhACT* as internal control gene.

Primers for circadian clock genes were designed using pcrEfficiency [35] (Appendix A) and the following protocol was used for 40 cycles: 95 °C for 5 s, 60 °C for 20 s and 72 °C for 15 s. Samples were run in duplicate. Primer combinations were tested with genomic DNA from Mitchell, and we found that all of them gave a single copy DNA on agarose gels. The endpoint PCR was further verified by melting point analysis where all primer combinations gave a single peak of melting (Appendix A).

### 2.5. Image Acquisition

We used an image acquisition system described previously [36]. Plants were grown inside a growth chamber comprising LED lights covering from UV to red light. Day and night images were taken by activating an Infrared light at 840 nm wavelength during short intervals of time (3 s). Images were acquired every ten minutes with an artificial vision camera comprising two CCD sensors, a multichannel 24-bit RGB absorbing at 610 nm, 540 nm and 460 nm, and a monochromatic sensor capturing at 800 nm. The acquired images have a resolution of 1296 × 966 pixels.

We obtained data using transgenic lines and compared them to the segregating siblings. Leaf growth and movement in seedlings was recorded for a period of 12 days with a total of 1728 images. Stem and flower growth was recorded for a period of 5 days and 16 h for line 3 and 3 days for line 10, for a total of 822 images and 432 images respectively. Using a semi-automatic procedure, we measured the length and angle of the longitudinal axis as referred to the horizontal plane of the flowers, as well as the length of the stem. The acquisition intervals were 1 hour for line 3 and 2 h for line 10, to explore differences in flower and stem growth patterns between wild-type and transgenic individuals. Analysis of growth was performed using the R package grofit [37]. Graphics were done using the color-blind friendly palette.

### 2.6. Scent Analysis

Flowers were placed in a glass beaker with a solution 4% of glucose inside a desiccator. Emitted volatiles were collected with twisters from the headspace every three h and analyzed by GM/CS as described [16]. Total and relative amounts were calculated using total integrated area divided by fresh weight as described before [38]. Detection of rhythmic scent emission was performed using the JTK_CYCLE algorithm [39] implemented in the R package MetaCycle [40]. Scent profile figures were plotted using the R library ggplot2 [41], using color palettes providing by the package viridis [42].

## 3. Results

### 3.1. The Petunia Genomes Have a Single ZTL Gene

We mined the petunia genomes to identify ZEITLUPE/FLAVIN-BINDING KELCH REPEAT F-BOX (ZTL/FKF) orthologs and paralogs using BLAST. We found two genes with homology to the Arabidopsis genome in *P. axillaris*: Peaxi162Scf01124g00126.1 and Peaxi162Scf00655g00114.1; and three in *P. inflata*: Peinf101Scf01230g02037.1, Peinf101Scf02808g00015.1 and Peinf101Scf04186g00007.1 (Appendix A).

We performed a phylogenetic reconstruction of the genes found in petunia and compared them to the LOV-F-box-KELCH proteins from several monocots, dicots and the basal plant *Marchantia polymorpha* (Appendix A). As previously mentioned, the silencing of *PhCHL* caused a major change in the composition of floral scent. Thus, we named the gene CHANEL (*PhCHL*) (see below).

The phylogenetic reconstruction shows two major clades, one containing all the FKF)-like genes and a second one comprising the ZTL and the Arabidopsis LOV KELCH PROTEIN2 (LKP2). In both clades, two subclades separate monocots from dicots. The Marchantia FKF like gene [43] falls somewhere between the FKF group of coding genes and the ZTL coding set of genes. The genomes of *P. axillaris* and *P. inflata* appear to have one gene corresponding to *PhCHL*, but differ in the copy number of FKF that is found in two copies in *P. inflata*. According to the ancestral region by gene analysis of *Petunia hybrida* W115, the ZTL-FKF genes present in *P. hybrida* correspond to *P. axillaris* [19]. Thus, based on the phylogenetic reconstruction, we identified Peaxi162Scf01124g00126.1 as *PhCHL* and Peaxi162Scf00655g00114.1 as *PhFKF*.

### 3.2. The Expression of PhCHL is Organ Specific and Affected by Day Length 

The *PhCHL* expression has been studied in leaves and seedlings of *Nicotiana attenuata* and Arabidopsis [17,44,45]. In these plants and tissues, *PhCHL* does not show rhythmic expression. We determined the set of reference genes for circadian expression in leaves and petals in order to establish the circadian gene expression patterns in petunia (see Materials and Methods). We analyzed the expression of *PhCHL* in petunia corolla using a 12:12 LD light regime and found that the expression had its maximum at ZT9 (Figure 1A). We found that the expression at ZT9 was significantly higher, and the lowest expression occurred at ZT15. Although the expression of *PhCHL* in corollas showed a peak at ZT9, the mathematical analysis for circadian oscillation using the JTK_CYCLE algorithm indicated that the expression of *PhCHL* in petunia corollas was not rhythmic (*p* = 1). We also analyzed the expression of *PhCHL* in leaves and the pattern was similar to corollas: *PhCHL* increased at ZT9 (Figure 1A), and its expression was not rhythmic (*p* = 1). Under short days (8L:16D), the expression of *PhCHL* in leaves reached its maximum at midnight (ZT17) while the maximum expression in petals was detected during the afternoon (ZT5) (Figure 1B). As observed under 12L:12D, *PhCHL* did not display a rhythmic expression either in leaves or petals (*p* > 0.05).

### 3.3. Silencing PhCHL Does Not Affect PhFKF

We knocked down *PhCHL* using a RNAi hairpin construct and found significant down regulation of ZTL in several lines. We used two lines for further analysis. *RNAi:PhCHL10* showed a down regulation of 80% (*p* = 0.032), while RNAi:PhCHL3 showed a downregulation of 68% (*p* = 0.047) (Figure 2A). We also analyzed the expression of FLAVIN-BINDING KELCH REPEAT F-BOX 1 (PhFKF), to discard co-silencing of the paralogous genes (ZT5). The expression of PhFKF did not differ significantly between transgenic and non-transgenic petunias (Figure 2B).

### 3.4. PhCHL Is a Positive Regulator of Lateral Organ Growth 

We quantified the effect of *RNAi:PhCHL* on plant development and found that while not all parameters analyzed showed statistically significant changes, in general terms, *PhCHL* appears to play a role in the production of above ground biomass affecting organ size (Table 1). Thus, *RNAi:PhCHL* plants were significantly shorter than the non-transgenic siblings by roughly 23%. We analyzed floral size by measuring the length of the tube and the maximum expansion of the limb and found that, while the tube length was not always affected, the limb expansion was significantly reduced by *RNAi:PhCHL*, with size changes in the range of 12–18% (Table 1). Finally, we did not find differences in chlorophyll content (Table 1) or in the general canopy architecture i.e., number of branches (data not shown).

We can conclude that *RNAi:PhCHL* is a positive regulator of lateral organ size with a pleiotropic effect causing a decrease in shoot and flower size.

### 3.5. PhCHL Is Involved in Flower Angle Changes but Not Leaf Movement

As size was negatively affected by *RNAi:PhCHL* (Table 1), we used a phenomics approach with time lapse image acquisition to identify the effects on growth kinetics [46]. We tried to analyze leaf growth in seedlings; however, leaf movement, a trait under light and circadian control, was so extreme that we could not obtain reliable data for growth (Appendix A). Changes in leaf position appeared to correspond to day/night changes. We measured the changes in leaf position every hour for a period of four days. We found that both non-transgenic and transgenic lines had open leaves during the day and closed leaves during the night (Appendix A). The changes in leaf opening and closing, in terms of speed or timing were not affected by the *PhCHL* expression levels (Figure 3A). Indeed, a mathematical analysis of the period, lag phase and amplitude did not show significant differences between *RNAi:PhCHL* and wild type (Appendix A).

We analyzed flower angle against the horizontal axis, as it is a parameter related to pollination that is regulated by *NaZTL* in *Nicotiana attenuata* [17]. Both wild-type and transgenic lines had a similar daily pattern of flower movement; however, RNAi line 3 flowers had always higher angles than the non-transgenic siblings. A decomposed time series showed that their daily changes in angle were slightly advanced compared to wild type (Figure 3B).

### 3.6. Differential Effect of PhCHL on Stem and Flower Growth Rate

The difference in overall plant size indicated a role of *PhCHL* in plant growth. We analyzed stem growth and found that growth curves (Table 2) were similar in wild-type and transgenic plants (Figure 4A; Table 2). Still, maximal growth speed in transgenic line 3 was 80.7% compared to wild type and area under the curve was 82%. The data indicates that stems are shorter because growth occurs during shorter periods of time, but growth speed appears to be less affected in the stem (see discussion).

We analyzed the increase in flower length. Down regulation of *PhCHL* caused a strong decrease in flower growth speed in line 3 (Appendix A). The wild-type flowers had an overall growth speed that was always higher than in transgenic lines (Figure 4B). We calculated the differences in growth speed and found that the maximal slope indicating growth speed was 1.4-fold higher in wild type compared to transgenic line 3 (Table 2). We analyzed line 10, but due to flowering time differences, we were able to obtain data only from older flowers. These flowers also grew at a lower speed than wild type, but differences were not so pronounced (Appendix A) (Figure 4C).

Growth of hypocotyls in Arabidopsis is gated by the plant circadian clock [47]. We tested the hypothesis that petunia flowers may grow at different speeds during day and night. Indeed, we found that overall flower growth speed was higher during the day than during the night, coming to a sharp decrease in growth speed when flowers open (Figure 4D). The flowers corresponding to *RNAi:PhCHL* always grew slower than the wild types, but the time required to become fully developed was not affected by *PhCHL* (Figure 4D). Altogether, our data shows that *PhCHL* plays a role in organ growth speed and duration that is organ-specific.

### 3.7. PhCHL Coordinates Daily Changes in Scent Profiles 

Petunia flowers emit mostly benzenoids/phenylpropanoids, including methyl benzoate, benzaldehyde and benzyl benzoate [48]. We also identified emission of terpenoids, cadinene and limonene. We selected 17 volatile organic compounds (VOCs) to analyze the effect of *PhCHL* silencing in volatile emission for a 24-hour period (Table 3). The major VOC emitted by P. hybrida was methyl benzoate (74.90% of selected volatiles) whereas cadinene was the volatiles with the lowest emission (0.02% of selected compounds). Methyl benzoate was also the principal emitted volatile in transgenic lines: 79.1% in *RNAi:PhCHL3* and 58.51% in *RNAi:PhCHL10* petunia flowers. The volatile with the lowest emission was acetophenone for *RNAi:PhCHL3* (0.01%) and cadinene for *RNAi:PhCHL10* (0.018%). Due to the high emission of methyl benzoate in all petunia lines, we analyzed the volatile profile excluding this compound (Figure 5, Appendix A).

Excluding methylbenzoate (Appendix A), the second-highest emitted volatile was ethyl benzoate in wild-type petunia and *RNAi:PhCHL10* line (45.72% and 48.61% of selected compounds, respectively). In contrast, the emission of ethyl benzoate was lower in *RNAi:PhCHL3* (10.23%) and the second major volatile in this transgenic line was isoeugenol (24.38%). Isoeugenol represented 15.84% in wild type and 10.92% in *RNAi:PhCHL10* (Figure 5A). We found that certain volatiles increased their emission in both *RNAi:PhCHL* lines: benzaldehyde, phenylacetaldehyde, benzyl tiglate and benzyl 2-methylbutyrate (Figure 5A). The volatiles limonene, benzyl benzoate and acetophenone decreased their emission in *RNAi:PhCHL3* and *RNAi:PhCHL10* petunia flowers (Figure 5A) Finally, the quantities of some emitted volatiles varied among lines; the emission of benzyl acetate, cadinene, methyl salicylate, phenethyl acetate and phenethyl alcohol was higher in *RNAi:PhCHL3*, whereas the quantities of these compounds decreased in the *RNAi:PhCHL10* line (Figure 5A).

As the quantitative changes in scent could occur at different times of the day, we collected scent every three h in an 8L:16D cycle. The profile analysis of floral compounds throughout 24 h revealed modifications in the contribution of single compounds to the scent profile.

Methyl benzoate was the major emitted compound by non-transgenic and transgenic petunia flowers. Its contribution to the floral scent varied from 69 to 79% in wild-type petunias, (Appendix A), from 73 to 87% in RNAi line 3 (Appendix A) and from 46% to 71% in RNAi line 10 (Appendix A). Methyl benzoate tended to decrease during the light period in wild-type and *RNAi:PhCHL10* plants, in contrast in *RNAi:PhCHL3* the highest contribution to the floral scent occurred at ZT5 (Appendix A).

As mentioned above, we excluded methyl benzoate to analyze the temporal variation in scent emission profiles of those volatiles with lower emission. Briefly, in the wild-type plants, benzyl 2-methylbutyrate contribution to the scent profile increased during the light period (ZT2, ZT5). Cadinene, had the highest contribution at dawn and transition to day light (ZT23-ZT2). In contrast, the VOCs benzaldehyde, ethyl benzoate, limonene and phenylacetaldehyde showed their major contribution to the scent composition at dusk and early night (ZT8, ZT11). Acetophenone and benzyl benzoate displayed their highest contribution to the flower aroma at midnight (ZT14, ZT17). Finally, benzyl alcohol, benzyl acetate, eugenol, isoeugenol, methyl salicylate, phenethyl alcohol and phenethyl acetate percent composition was higher at late night (ZT20, ZT23) (Figure 5B).

When we analyzed the floral scent composition in RNAi petunias, three types of changes in the scent profile were observed. A set of volatiles showed similar profiles to the wild type (Figure 5B–D). These included of benzaldehyde, benzyl 2-methylbutyrate, cadinene, eugenol, isoeugenol and phenethyl alcohol. A second set of volatiles showed increased production during th late night and dawn (ZT23) as in wild type but did not decay during the light period (ZT2). Finally, acetophenone that had its maximal emission at midnight (ZT14–ZT17) and methyl salicylate (ZT20–ZT23) shifted their maxima to the light period at ZT2–ZT5.

These changes in timing of emission are reflected in the different quantitative combinations of VOCs found between wild-type and transgenic lines. Indeed, at the time when lights went off at ZT8, ethyl benzoate was the second important emitted volatile in W115 (70% of selected volatiles), decreasing in RNAi:PhCHL10 to 57% and RNAi:PhCHL3 to 8.7%. The VOCs acetophenone and phenenthyl alcohol also decreased their emission in both transgenic lines. In contrast, benzaldehyde emission increased in both transgenic lines from 6.85% W115 (6.85%) to 41.8% in RNAi:PhCHL3 and 22.1% t in RNAi:PhCHL10. Phenylacetaldehyde also increased its emission in both RNAi:PhCHL3 (18.7%) and *RNAi:PhCHL10* (4.3%) as petunia this volatile only represented a 1.4% of the profile in the wild-type (Figure 5B–D). When the lights went on at ZT0, scent also differed in transgenic and non-transgenic petunia flowers. The predominant compounds in the wild-type scent profile were phenethyl alcohol (30.5%), benzyl alcohol (21.9%) and isoeugenol (19.3%). In contrast, the major compounds in *RNAi:PhCHL3* were isoeugenol (33%), phenethyl alcohol (21.8%) ethyl benzoate (12.7%) meanwhile in *RNAi:PhCHL10* were ethyl benzoate (62.7%), phenethyl alcohol (11.4%) and isoeugenol (9.6%) (Figure 5B–D).

### 3.8. PhCHL Is Required for Timing of Scent Profiles

Petunia is considered a nocturnal plant; its scent emission is synchronized with pollinator activity. We analyzed the circadian rhythmicity of scent emission and the effect of silencing the gene *PhCHL*. In wild-type plants, benzaldehyde, benzyl acetate, benzyl alcohol, benzyl benzoate, eugenol, isoeugenol, methyl benzoate, phenylacetaldehyde, phenethyl acetate and phenethyl alcohol were rhythmically emitted (*p* < 0.05), while acetophenone, benzyl 2-methylbutyrate, benzyl tiglate, cadinene, limonene, ethyl benzoate and methyl salicylate turned out to be arrhythmic (*p* > 0.05) (Table 4). In the RNAi plants, most compounds emitted in a rhythmic fashion in wild type were emitted in a rhythmic manner (Table 4). Moreover, benzyl tiglate, benzyl 2-methylbutyrate and methyl salicylate were not rhythmically emitted both in W115 and *RNAi:PhCHL10* line oscillated significantly in *RNAi:PhCHL3* petunias (Table 4).

We also analyzed the phase or time point with the highest emission of selected volatiles. Most compounds increased their emission during the dark period, and we classified the volatiles according to their maximum emission. In the wild-type petunia, first group comprised benzyl tiglate and cadinene, which showed their peaks during the subjective day (ZT2 and ZT5, respectively). The second group included the volatiles ethyl benzoate and acetophenone, that reached its maximum emission at early night (ZT11 and ZT12.5, respectively). The third group consisted of those VOCs which peaked at midnight, from ZT15 to ZT18.5: benzaldehyde, benzyl acetate, limonene, isoeugenol, methyl benzoate and phenylacetaldehyde. Finally, the fourth group included benzyl 2-methylbutyrate, benzyl acetate, benzyl alcohol, eugeno, methyl salicylate, phenethyl acetate and phenethyl alcohol, that showed their maximum emission at late night (ZT20 to ZT23) (Table 4).

Analyzing the phase of emitted volatiles in transgenic lines revealed important differences. We found four types of behavior. First, the volatiles benzyl acetate, eugenol and phenethyl acetate did not change their peak in wild-type and transgenic petunias (Table 4). Second, benzyl benzoate, cadinene and ethyl benzoate delayed their maximum emission in both transgenic lines (Table 4). Third, acetophenone, benzyl alcohol and phenethyl alcohol displayed an advanced phase in *RNAi:PhCHL* plants (Table 4). Finally, a group of volatiles did not follow the same pattern in *RNAi:PhCHL3* and *RNAi:PhCHL10* lines; benzyl 2-methylbutirate and isoeugenol did not change their peak in *RNAi:PhCHL3* compared to W115 petunias whereas these compounds were emitted in advanced in the line *RNAi:PhCHL10*; on the other hand, the compounds benzaldehyde and methyl benzoate peaked at the same time in non-transgenic and *RNAi:PhCHL10* petunias but in the *RNAi:PhCHL3* line this peak was advanced (Table 4).

Altogether, *PhCHL* plays a fundamental role in coordinating floral scent profile of petunia that could be related to the timing of maximal production of different VOCs. The effect on single compounds appears to be opposite in some cases, suggesting a control of scent pathway at one of several points that we do not understand yet.

## 4. Discussion

The *ZTL* gene in Arabidopsis plays a central role in the circadian clock regulation and affects several traits such as flowering time [45], and in *Nicotiana* it also controls flower daily movement and the emission of benzyl acetone [17]. Here we have performed a systematic study of *PhCHL*, the ortholog of *ZTL*, and its outputs in *Petunia hybrida*, uncovering several unreported functions.

The ZTL/FKF/LKP2 proteins are blue light receptors in Arabidopsis. The expression of *ZTL* is not rhythmic in Arabidopsis or *Nicotiana* [44,45]. While *PhCHL* showed a clear peak of expression in 12L:12D, we could not find a significant rhythm, indicating a conserved expression pattern with other plants where it has been analyzed. This is the first time that *PhCHL*, a *ZTL* ortholog, has been analyzed in petals. The similarity of expression of *PhCHL* in leaves and petals indicates that the leaf and petal clock may be partly conserved. However, this needs further analysis with more genes, as clock transcriptional structure has been shown to change in roots, pods or seeds [49,50].

One of the classical experiments in Chronobiology in plants was performed by d’Ortous de Mairan. He demonstrated that leaf movement was rhythmic and endogenous [51]. Leaf angle and leaf movement have been widely studied in crops, including soybean and maize [52,53] and in the canonical model Arabidopsis. The analysis of circadian clock mutants in Arabidopsis revealed that the clock genes *GIGANTEA* (*GI*) and *EARLY FLOWERING 4* (*ELF4*) are required to maintain leaf movement rhythmicity [54,55]. In petunia, we detected periodic changes in leaf position but diel changes in position were not affected by the down-regulation of *PhCHL*. This result suggests that *PhCHL* did not play a critical role in leaf rhythms at early stages of development.

The first apparent effect of down-regulating *PhCHL* was a decrease in plant body size. A detailed analysis using time lapse images showed that the stem of wild-type plants and siblings knocked down for *PhCHL* grew at similar speed. This suggests that the identified decrease in shoot length maybe the result of smaller primordia, decreased growth duration or a combination of both phenomena. In contrast, petal development appeared to be affected in growth rate. Furthermore, petal growth occurred at higher rates during the day than during the night both in wild-type and transgenic plants, suggesting that gated growth of flowers was independent of *PhCHL*. Work in Arabidopsis and maize has shown that the process of growth rate and duration maybe differentially affected by mutations [56,57,58]. Our results show that measuring two different organs may yield different results related to growth rate and duration. The plant circadian clock appears to have an organ specific resetting in roots, pods and seeds [50,59,60,61]. In this context, shoot growth is driven by the shoot apical meristem while flowers have undergone major organ identity reprogramming. As both cell division and expansion are under direct control of the clock [62,63], our results indicate a somewhat different interpretation of loss of *PhCHL* in stems and flowers.

We performed a complete scent profile analysis in wild-type and knockdown lines. Floral scent comprised 98.66% benzenoids/phenylpropanoids and 1.34% terpenoids in wild-type petunias. This composition was somewhat similar in transgenic lines, indicating that the effects of *PhCHL* are not specific for a single volatile family of compounds. The emission of several phenylpropanoids/benzenoids compounds, such as methyl benzoate, benzyl alcohol, benzyl benzoate o benzaldehyde, displayed a rhythmic oscillation whereas acetophenone or ethyl benzoate (phenylpropranoids/benzenoids) and the terpenoids cadinene and limonene did not. In addition, the amount of emitted volatiles tended to increase and peak during the dark phase, coinciding with petunia pollinators activity [64]. These results are similar to those described previously for whole flowers, indicating that petal volatiles and those emitted by other organs have similar control mechanism [14]. This suggests that the circadian clock plays a key role in the regulation of biosynthesis and emission of certain compounds that interacts and/or attracts pollinators. In contrast, volatiles which did not oscillate such as acetophenone, or terpenoids may play a role in defense [65,66].

The disruption of *PhCHL* shifted the maximum emission of benzyl alcohol was advanced whereas the peaks of benzyl benzoate and ethyl benzoate were delayed. Other volatiles, such as benzyl alcohol, eugenol and phenethyl acetate, were not affected by the knockdown of *PhCHL*. These results suggest that *PhCHL* was involved, directly or indirectly, in the emission pattern of certain volatiles. However complete knockouts for *PhCHL* may show a stronger effect in the volatiles affected and/or additional effects on those that appear to remain stable.

Flower fragrances play a complex biological role. The dominant nocturnal emitted volatiles of *Petunia axillaris*, benzaldehyde, benzyl alcohol and methyl benzoate, act as attractant of its nocturnal pollinator [64] whereas benzaldehyde has been described as a mild repellent [67]. On the other hand, the highest emitted compound by *P. integrifolia* is benzaldehyde [64]. In *Petunia hybrida*, the three major compounds were methyl benzoate, ethyl benzoate and isoeugenol. In contrast, the three principal released VOCs in *RNAi:PhCHL3* were methyl benzoate, isoeugenol and benzaldehyde, while in *RNAi:PhCHL10*, they were methyl benzoate, ethyl benzoate and benzaldehyde. In addition, in the present work, we described how the proportion of the compounds emitted by wild-type and *RNAi:PhCHL* petunias changed throughout a 24 h period. Previous studies that covered the down-regulation of the clock genes *LHY* and *ZTL* in petunia and wild tobacco, have report a reduction in volatile emission and production [17,18]. Our results showed that silencing *PhCHL* modified blend ratios, resulting in a different scent profile. Changes in emission pattern and fragrance composition may have an effect in pollinator attraction, plant defense against herbivores and pathogens or plant-plant signaling [68,69,70]. Our work shows that *PhCHL* plays a major role in the quantities and timing of VOC emission, thus coordinating the proper composition and daily changes of scent blends.

## Figures and Tables

**Figure 1 cells-08-00343-f001:**
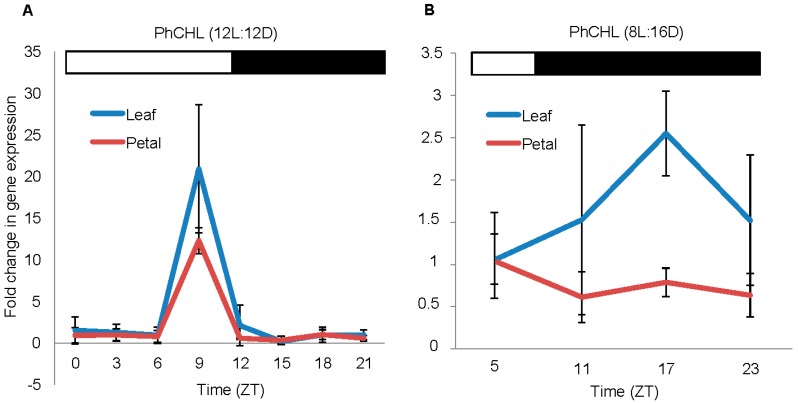
*PhCHL* expression in W115 petunia under 12LD (**A**) and 8L16D cycle (**B**) in leaves (blue line) and petals (red line) revealed a different pattern depending on light conditions. The maximum expression ocurred before the dark period (ZT9) in leaves and petals under a 12LD cycle (**A**). This maximum was delayed in leaves under short days. *PhCHL* maintained its peak before dusk in petals (**B**). Results represent average ± standard deviation from three replicates. White bar indicates the light period and black bar, the dark period.

**Figure 2 cells-08-00343-f002:**
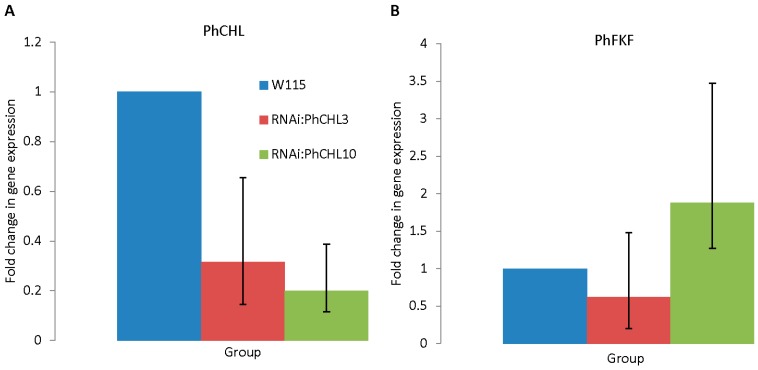
*PhCHL* (**A**) and *PhFKF* (**B**) expression in wild-type (blue bar) and *RNAi:PhCHL* plants (red bar and green bar) petals. *PhCHL* expression was significantly down-regulated in *RNAi:PhCHL3* (*p* = 0.047) and *RNAi:PhCHL10* (*p* = 0.032) compared to the wild-type (**A**). Silencing *PhCHL* did not affect *PhFKF* expression *RNAi:PhCHL3* (*p* value = 0.6), *RNAi:PhCHL10* (*p* value = 0.06) (**B**).

**Figure 3 cells-08-00343-f003:**
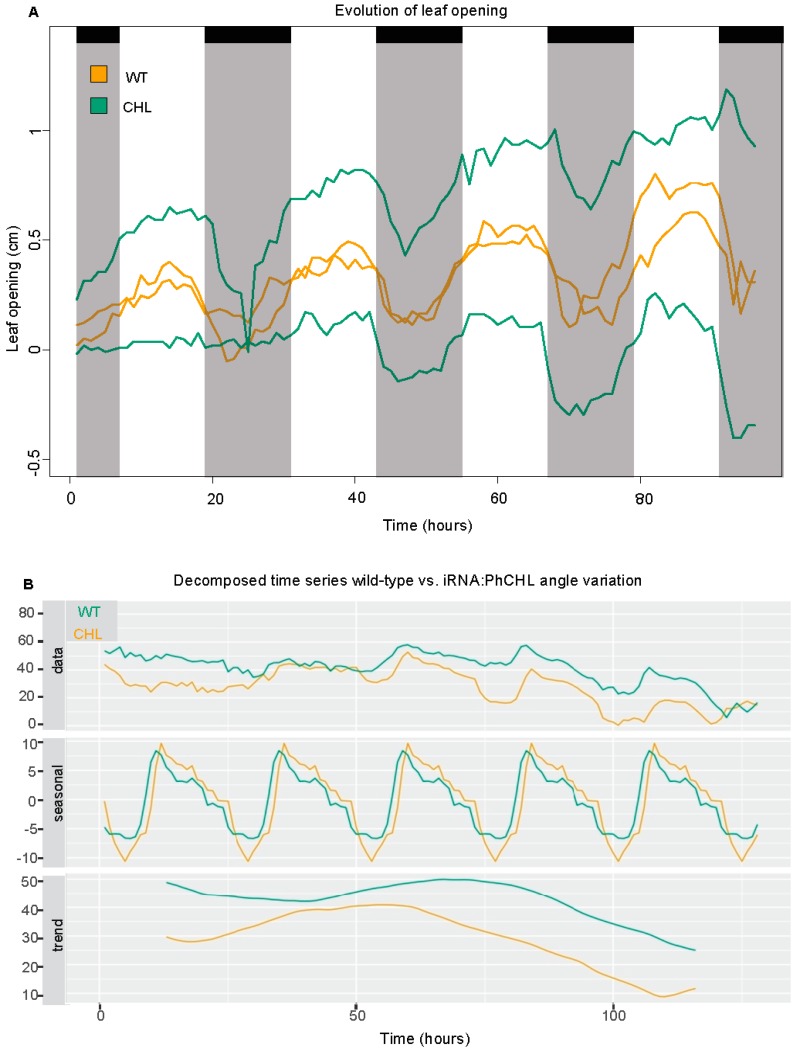
Down regulation of *PhCHL* does not affect leaf movement (**A**) The distance between the center of the plant and the end of each leaf is shown. Furthermore, these data correspond to the distance in the X axis. Negative values correspond to leaf positions where the tip of the leaf passed the center of the plant. Down regulation of *PhCHL* causes increased flower angles and advanced daily changes (green line, **B**). Orange line represents WT plants, green line represents *RNAi:PhCHL* plants.

**Figure 4 cells-08-00343-f004:**
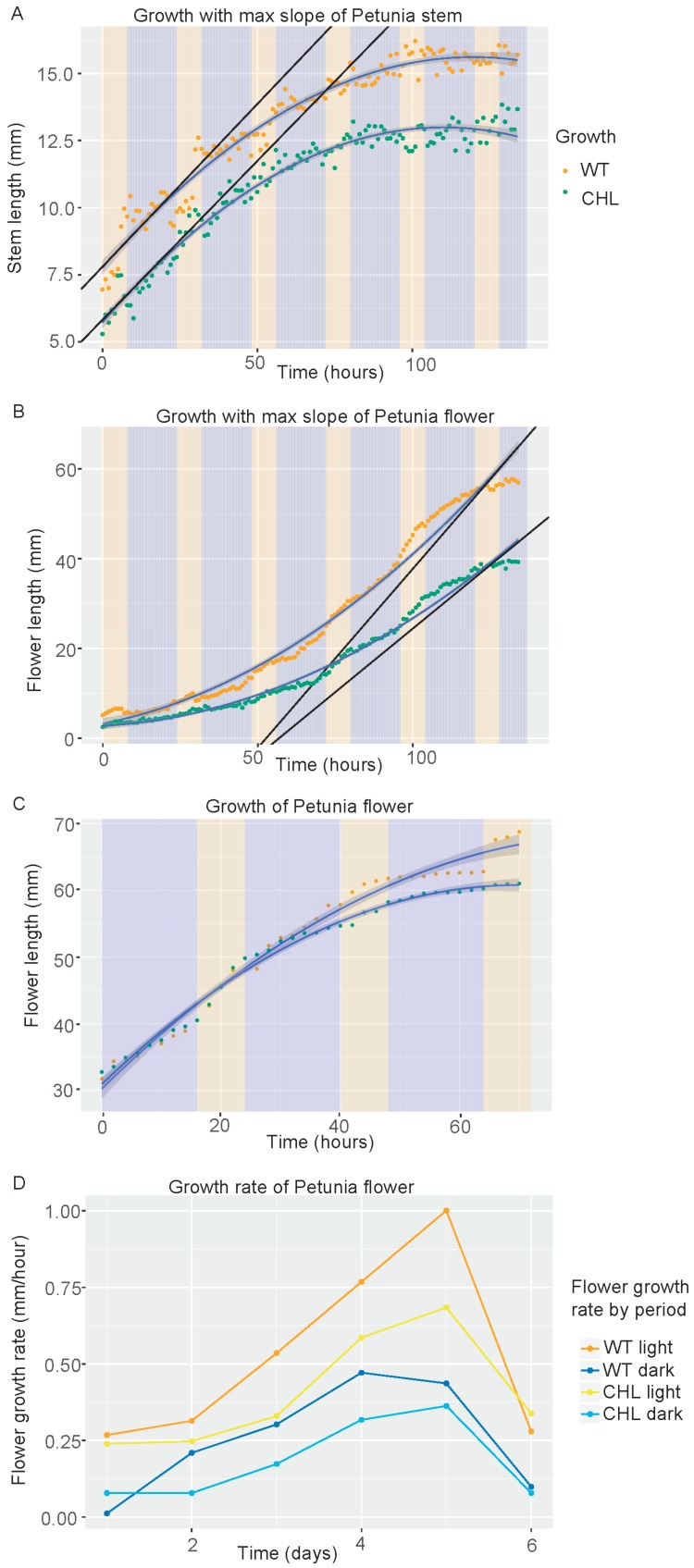
Down regulation of *PhCHL* affects growth length in stems (**A**) and growth rate in flowers (**B**,**C**). Graphics show the raw data (points), the adjusted curve and confidence interval (grey shade). Orange line represents WT plants, green line represents *RNAi:PhCHL* plants. Yellow and blue areas indicated light and dark period, respectively. (**D**) Flower growth rate by period, day versus night in wild-type and *RNAi:PhCHL* transgenic lines.

**Figure 5 cells-08-00343-f005:**
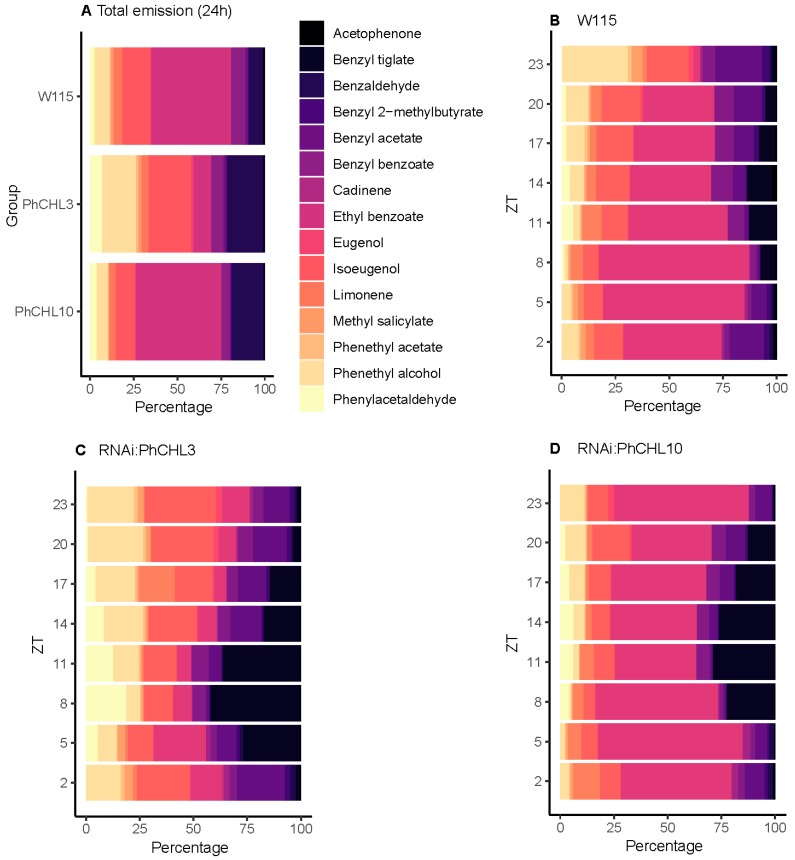
Petunia scent profiles expressed as percentage. Due to its high emission, methyl benzoate was excluded to visualize the remaining volatiles. Profiles were determined for 24 h emission under a 8L:16D cycle for the wild-type (first bar) and the transgenic lines (second and third bar) (**A**) and in a time-course (**B**–**D**), where each bar represent one sampling point (in ZT h) for wild-type (**B**), *RNAi:PhCHL3* (**C**) and *RNAi:PhCHL10* (**D**) petunia flowers.

**Table 1 cells-08-00343-t001:** Phenotypic analysis of *RNAi:PhCHL* lines 3 and 10 and comparison to non-transgenic siblings in T2 generation. Plant height, floral limb and floral tube length were measured from *RNAi:PhCHL* lines (3, 10) and the wild-type W115. Results, in centimeters, represent average ± standard deviation. *p*-value < 0.05 indicates a significant difference between non-transgenic and transgenic plants, significance levels are indicated with one asterisks (* for *p* < 0.05), two asterisks (** for *p* < 0.001) and three asterisks (*** for *p* < 0.0001). NS indicates not significant differences.

Measure	RNAi:PhCHL		W115	*p*-Value	
	Line 3	Line 10		RNAi:PhCHL3	RNAi:PhCHL10
Plant height	25.1 ± 1.89	21.7 ± 4.18	37.33 ± 3.0	0.07	0.022 *
Floral limb	4.37 ± 0.44	4.76 ± 0.55	5.38 ± 0.29	0.00009 ***	0.0038 **
Floral tube	4.07 ± 0.37	4.81 ± 0.51	4.82 ± 0.35	0.00055 ***	NS
Chlorophyll	39.43 ± 6.32	36.65 ± 9.77	39.30 ± 9.80	NS	NS

**Table 2 cells-08-00343-t002:** Growth rate of petunia stems and flowers. Maximal growth is expressed in mm. Stems grew from day 0 and total growth refers to size achieved after 5 days and 16 h taking as reference the stem at time zero. Maximum growth for flowers refers to the same period but comprises the overall floral size. The maximum slope depicts growth rate while the area under the model (integral) gives an estimation of the overall difference in accumulated growth.

Group	Rate	Max.Growth/Std.Error	Max.Slope/Std.Error	Area under Model
WT	Stem growth	16.084/0.142	0.121/0.005	1778.3
*RNAi:PhCHL*	Stem growth	12.984/0.096	0.118/0.006	1471.53
WT	Flower growth	57.698/0.403	0.798/0.013	3598.88
*RNAi:PhCHL*	Flower growth	39.677/0.344	0.566/0.011	2328.95

**Table 3 cells-08-00343-t003:** Retention time (RT) expressed in minutes, name and CAS number (assigned by the Chemical Abstract Services) of selected volatiles.

RT	Name	CAS
4.873	Benzaldehyde	100-52-7
6.435	Limonene	138-86-3
6.539	Benzyl alcohol	100-51-6
6.735	Phenylacetaldehyde	122-78-1
7.230	Acetophenone	98-86-2
7.825	Methyl benzoate	93-58-3
8.149	Phenylethyl alcohol	60-12-8
9.125	Benzyl acetate	140-11-4
9.244	Ethyl benzoate	93-89-0
9.668	Methyl salicylate	119-36-8
10.720	Phenylethyl acetate	103-45-7
12.287	Eugenol	97-53-0
12.711	Benzyl 2-methylbutyrate	56423-40-6
13.587	Isoeugenol (isomers)	97-45-1; 5932-68-3
14.254	Benzyl tiglate	37526-88-8
14.625	Cadinene	483-76-1
17.525	Benzyl benzoate	120-51-4

**Table 4 cells-08-00343-t004:** Analysis of emitted volatiles with MetaCycle. The JTK_CYCLE algorithm implemented in the R library “MetaCycle” was used to detect rhythms in emitted volatiles of petunia wild-type (W115) and transgenic *RNAi:PhCHL* lines (*PhCHL3*, *PhCHL10*). Volatiles with significant *p* value (*p* value < 0.05) showed a rhythmic emission. Phase is defined as the time point, in ZT h, with the highest emission.

Volatile	W115 *p* Value	Phase	PhCHL3 *p* Value	Phase	PhCHL10 *p* Value	Phase
Benzyl tiglate	1.000	2	0.014	0.5	0.476	5
Cadinene	0.445	5	0.246	23	1.000	9.5
Ethyl benzoate	1.000	11	1.000	20	1.000	20
Acetophenone	1.000	12.5	1.000	8	0.210	11
Benzaldehyde	0.000	15.5	0.000	12.5	0.000	15.5
Benzyl benzoate	0.001	15.5	0.023	17	0.001	17
Limonene	0.165	15.5	0.280	17	1.000	11
Phenylacetaldehyde	0.001	15.5	0.000	12.5	0.000	20
Methyl benzoate	0.017	17	0.003	15.5	0.011	17
Isoeugenol	0.001	18.5	0.007	18.5	0.014	17
Benzyl acetate	0.003	20	0.001	20	0.005	20
Eugenol	0.001	20	0.000	20	0.019	20
Methyl salicylate	1.000	20	0.009	20	1.000	15.5
Phenylethyl acetate	0.048	20	0.002	20	0.000	20
Phenylethyl alcohol	0.000	20	0.000	18.5	0.000	15.5
Benzyl alcohol	0.000	21.5	0.007	20	0.000	20
Benzyl 2-methylbutyrate	1.000	23	0.011	23	0.194	2

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
