# Peer review of "The Petunia CHANEL Gene is a ZEITLUPE Ortholog Coordinating Growth and Scent Profiles"

_cells, 2019, doi:10.3390/cells8040343_

Reviewer 1 Report

The manuscript submitted by Terry et al., reports the involvement of the photoreceptor CHANEL, the ZEITLUPE ortholog in petunia, in the coordination of the flower growth and the scent chemical composition.

The combination of experimental approaches, genomic, plant imaging and metabolic analysis help to support the conslusion.

Although minor revisions have to be incorporated in the manuscript most of it is cleraly written. Therefore, the manuscript by Terry et al. is suitable for publication in Cells once the minor revisions detailed below are corrected.

Minor revision:

-in the M&M section, th eauthors should better describe the promoter used to drive the expression of the RNAi:PhCHL construct.

-In general, the figures are of poor quality, for example figures 2, 3, 6, 7. The figure's quality must be improved to reach the quality standard

-In the figure 2, the graph legend is incomplete. The correspondance of the light blue color is missing.

Author Response

Minor revision:

-in the M&M section, th eauthors should better describe the promoter used to drive the expression of the RNAi:PhCHL construct.

Comment #1 We have described the promoter with more details on the M&M part (line 104).

-In general, the figures are of poor quality, for example figures 2, 3, 6, 7. The figure's quality must be improved to reach the quality standard

Comment #2 We have improved the quality of figures 2, 3, 6 and 7.

-In the figure 2, the graph legend is incomplete. The correspondance of the light blue color is missing.

Comment #3 We have completed the graph legend in the figure 2.

Reviewer 2 Report

In this manuscript, Terry et al. studied Petunia CHANEL gene. They found it was a ZEITLUPE ortholog coordinating growth and scent profiles.

A lot of abbreviations were used without spelling out the full name or introduced, which may cause uncertainty and confusion. For example, FKF, VOC, ….

The way the results were presented should be improved. For example, Figure 1, Figure 2 and Figure 3, table 1 and Table 2, could be moved to supplementary data. The figure quality should be increased…..

Author Response

In this manuscript, Terry et al. studied Petunia CHANEL gene. They found it was a ZEITLUPE ortholog coordinating growth and scent profiles.

A lot of abbreviations were used without spelling out the full name or introduced, which may cause uncertainty and confusion. For example, FKF, VOC, ….

Comment #4 We have added the full name of genes and other abbreviations (lines 61, 67, 187, 198 and 336).

The way the results were presented should be improved. For example, Figure 1, Figure 2 and Figure 3, table 1 and Table 2, could be moved to supplementary data.

Comment #5 We have moved Figure 1, 2 and 3 together with Tables 1 and 2 to Supplementary data as suggested. As a result figure and table numbers along the manuscript have changed (lines 97, 137, 142, 154, 192, 195, 215, 236, 249, 275, 277, 294, 295, 331, 337 and 408).

The figure quality should be increased…..

Comment #6 We have improved the figure quality.

Round  2

Reviewer 2 Report

Please show all your revisions in other color. It is really confusing to see some revisions displayed and some not.

Please read through the manuscript carefully before resubmission. All Figures and Tables should be in order in text. For example, this version, Table S1 is the last. Table S2 shows first.

Author Response

Please show all your revisions in other color. It is really confusing to see some revisions displayed and some not.

Comment #1 We marked all the revisions in color in order to make them all more visible.

Please read through the manuscript carefully before resubmission. All Figures and Tables should be in order in text. For example, this version, Table S1 is the last. Table S2 shows first.

Comment #2 We have reviewed and corrected the order of figures and tables.